analytical chemistry/molecular biology

corneous beta-proteins, tortoiseshell, Cheloniidae, proteomics, combs, hawksbill turtle

**Author for correspondence:**
Caroline Solazzo
e-mail: solazzoc@si.edu

# Creation of a peptide database of corneous beta-proteins of marine turtles for the identification of tortoiseshell: archaeological combs as case study

## Caroline Solazzo[1], Jean Soulat[2] and Timothy Cleland[1]

[1]Smithsonian's Museum Conservation Institute, 4210 Silver Hill Road, Suitland, MD 20746, USA
[2]LandArc Laboratory, 5, rue Victor Chevin, 77920 Samois-sur-Seine, France

CS, 0000-0001-5092-0807; TC, 0000-0001-9198-2828

Tortoiseshell is a proteinaceous material derived from the scutes of marine turtles, and was shaped into an abundance of objects, especially luxurious items, at its peak in the seventeenth and eighteenth century. It has continued to be used even after the advent of plastics and remains one of the main causes of illegal poaching of marine turtles, in particular the hawksbill turtle *Eretmochelys imbricata*. Tortoiseshell is made of structural proteins, of which the most abundant are known as β-keratins, or 'corneous beta-proteins' (CBPs), a family of short proteins containing a central structure in β-sheets. There are, however, few CBP sequences of marine turtles in protein databases. The scutes of the five main species of marine turtles (*Chelonia mydas*, *Caretta caretta*, *Eretmochelys imbricata*, *Lepidochelys olivacea* and *Lepidochelys kempii*) were analysed by proteomics, using nano-liquid chromatography–Orbitrap–mass spectrometry to generate peptidic markers for species identification. A total of 187 marker sequences were identified, the large majority of them obtained from automated de novo sequencing. The sequences were classified into peptides A to F: A to D at the N-terminus and central region that forms the β-pleated sheets, E1–4 for a variable region of glycine-repeats region and F at the C-terminus. The markers were tested against a set of combs discovered in various archaeological sites of modern period in France, successfully identifying hawksbill turtle and highlighting patterns of degradation in archaeological tortoiseshell.

# 1. Introduction

## 1.1. Historical use and origin of tortoiseshell

Globally, marine turtles have been exploited for their bones, meat and their most prized appendage, the epithelial layer of the scutes that is commonly referred to as tortoiseshell [1]. It is obtained predominantly from the hawksbill turtle, for which the high demand has contributed to the major decline of the species [2]. In the past, it might also have been acquired from the green and loggerhead turtles [3–6] and even land tortoises in antiquity [7]. Tortoiseshell is the outer layer of either the carapace or plastron, peeled from the bony structure of the shell using heat (by boiling or over a fire) [3]. Working with hot salt water, the material was worked, flattened and moulded in the desired shape [8,9]. Historically sought after as an exotic material, tortoiseshell was used in a variety of small items such as combs, knife handles, fans, cases and other boxes, in bookbinding, and as inlays for furniture [5,6], and thus transformed into luxurious objects particularly in the seventeenth and eighteenth century [6,10]. Less frequently recovered from archaeological excavations, in part due to the fast decay of the tissue in the soil and possibly lack of proper identification, it has nonetheless been found in the form of scraps or ancient comb fragments in modern periods [11,12].

The trade of tortoiseshell is known from the first-century AD Greek treatise *Periplus Maris Erythrae* (The Periplus of the Erythraean Sea) that described the trade between the Mediterranean Coast to trading ports in India or Indonesia [13]. It probably started even earlier, and there is abundance of historical accounts that the trade was widespread in China and other parts of Asia [13]. Marine turtles have been exploited for thousands of years, possibly contributing to the important declines in population for all species seen today [14]. The International Union for Conservation of Nature (IUCN, https://www.iucnredlist.org/) lists all marine turtles as vulnerable (the loggerhead, *Caretta caretta* [15], and the Pacific ridley, *Lepidochelys olivacea* [16]), endangered (the green, *Chelonia mydas* [17]), or critically endangered (the hawksbill, *Eretmochelys imbricata* [18] and the Atlantic ridley *Lepidochelys kempii* [19]); the flatback, *Natator depressus*, has a Data Deficient status. Indeed, in spite of the ban on hawksbill turtle imposed by the Convention on International Trade of Endangered Species of Wild Fauna and Flora (CITES) and followed by most signatories countries in the 1970s and 1980s, the species has continued to be poached and there is still widespread illegal trade of tortoiseshell to make objects such as jewellery, in particular in Japan for the bekko industry [2,20].

## 1.2. The Caribbean trade of tortoiseshell

In this study, archaeological samples were provided to obtain data towards the study of trade and processing of tortoiseshell originating from the Caribbean, as part of the 'Transatlantic Connections' research programme (initiative of the LandArc Laboratory). While there is an abundance of turtle remains in the Caribbean from the second millennium BCE [21,22] and elsewhere in the world from at least 5000 BCE attesting of the use of marine turtles in the past, the archaeological evidence is essentially based on turtle bones found in connection with food consumption, funerary offerings, or worked into tools and other objects [7,21]. Tortoiseshell was used to make fishhooks in the seventeenth and eighteenth century in the Caribbean [21,23], but has only been identified in one pre-Hispanic object from Mexico (possibly Yucatan coast), a Maya mosaic mask from the thirteenth to fourteenth century AD [24]. However, tortoiseshell becomes widespread in craftmanship from the sixteenth century onward, and trade with European countries intensified to obtain the exotic material for a variety of applications from small object to marquetry.

For metropolitan France, this exotic material must have come directly from the large commercial ports on the Atlantic seaboard such as Nantes, La Rochelle or Bordeaux where merchant ships unloaded and sold their cargo bought in the Lesser Antilles in the French colonies. Several accounts by French chroniclers from the seventeenth to eighteenth centuries report that sea turtles, in particular hawksbill, green or loggerhead turtles [25–27], were very numerous and strongly represented in the West Indies, the Cayman Islands [25,28], Martinique or Guadeloupe [26]. These three species were large, in particular hawksbill turtles, the most common in the West Indies. They were widely exploited for their meat first, which resulted in overexploitation and increased commercialization of turtles linked to the increase in population during the colonial period. This animal was used both to feed the crews onboard vessels and the populations of the colony, free or servile. As the shells were not edible, but had an obvious financial interest, they were then sold raw to be used in tortoiseshell

crafts in the West Indies or in France [25]. The trade in tortoiseshell and the attraction for this exotic material led to the establishment of a supply network that facilitated its distribution in Europe. This commercial network was the same as for other luxury products from the West Indies (sugar, tobacco, etc.). In France, in the seventeenth century, turtle shells came directly from Martinique and Guadeloupe [26] and were traded between the natives and the French living in the colonies. Several reports of the export trade to France during the eighteenth century clearly indicate that the sea turtle called 'caret' (another name for the loggerhead turtle) was an exotic product commonly found in ship cargoes [29,30]. Although these records do not specify in what form the shells were transported, it appears that they were cut in the West Indies and stacked in barrels, thus saving as much space as possible in the holds. The capital gain was clearly quite significant between the purchase price in the colonies and the sale price in France because turtle shells, a common raw material in the West Indies, were part of luxury products in France, at least between the seventeenth century and the first half of the eighteenth century.

## 1.3. Methods of analysis of tortoiseshell

A variety of materials have been used to imitate tortoiseshell, such as paint and lacquers, resins, animal horn, gelatin films, casein formaldehyde and other synthetic and semisynthetic plastics [3–5,8,9,31,32]. Most current methods of analysis are limited to identifying true tortoiseshell from these imitations, and are typically based on spectroscopic methods such as Fourier transform Raman [33], and attenuated total reflection Fourier transform infrared (ATR-FTIR) and diffuse reflectance infrared Fourier transform (DRIFT) spectroscopies [34–37]. However, the exact taxonomy of the tortoiseshell sample cannot be achieved through these methods. Recent research has shown that biomolecular methods had potential to identify the turtle species after tortoiseshell was made into an object [12], and even the geographical origin of the turtle based on mitochondrial DNA haplotypes (Atlantic and Indo-Pacific stocks in hawksbill turtle [38]). Peptide mass fingerprinting was used in the preliminary study that compared ancient tortoiseshell fragments to green and hawksbill turtles [12] and was used to distinguish six species of marine turtles based on bone collagen's markers [22].

## 1.4. Corneous beta-proteins

The seven extant species of sea turtles are grouped into two families: the Dermochelidae family with one species *Dermochelys coriacea* (leatherback turtle), and the Cheloniidae family to whom belong *C. mydas*, *C. caretta*, *E. imbricata*, *L. kempii*, *L. olivacea* and *N. depressus*. All Cheloniidae species have a hard shell made from a dorsal carapace connected to a ventral plastron and structured in two layers: the bony plates and the integumental layer of horny scutes growing from the dermis and epidermis [39]. The leatherback turtle is a soft-shelled turtle with a leathery skin, having lost the hard cornified scutes of the shell [1,40]. The hard scutes result from the cornification of the upper layers of the epidermis, the stratum corneum, through accumulation of proteins commonly known as β-keratins. Researchers have recently proposed to rename these proteins 'corneous beta-proteins' (CBPs) to remove the misleading association with the α-keratins (or intermediate filament proteins, IFPs) as the two groups of proteins do not share ancestry (different gene families) or sequence similarity [41]. Instead, the CBPs have their origin in the epidermal differentiation complex (EDC); they are the main structural proteins of birds' beaks, claws and feathers, and reptile scales, but do not occur in mammals [42].

The CBPs intermingle with 'soft' IFPs (cytokeratins) in the lower layers of the epidermis of turtles; they are shorter proteins (usually less than 200 residues), about half to one-third the molecular weight of the IFPs. In the IFPs, the rod domain between the head and tail regions conforms into an α-helix; in the CBPs, a small central region conforms into four antiparallel β-sheets in turtles and assembles with another CBP's β-sheet region to form a dimer [40,43,44] (figure 1). While the epidermis of soft-shelled turtles has a higher ratio of cytokeratins [45,46], in hard-shelled turtles, the CBPs accumulate and predominate in the upper layers of the epidermis, polymerizing into β-filaments through head and tail interactions to form the corneous layer [40,42,43]. The higher level of expression of CBPs in the scutes indicates they are the main contributors towards the hardness of the shell [45–47]. The 34 amino acid residues central region involved in the β-sheet structure has high homology among sauropsids [43]; it is rich in proline that prompts the turns in the protein chain, and hydrophobic amino acids such as valine, leucine and isoleucine that allow packing of the β-sheets through hydrophobic interactions [42,43]. The N- and C-regions of the CBPs do not take part in the β-sheet conformation and instead form the inter-filamentous matrix [42]. A high cysteine content in

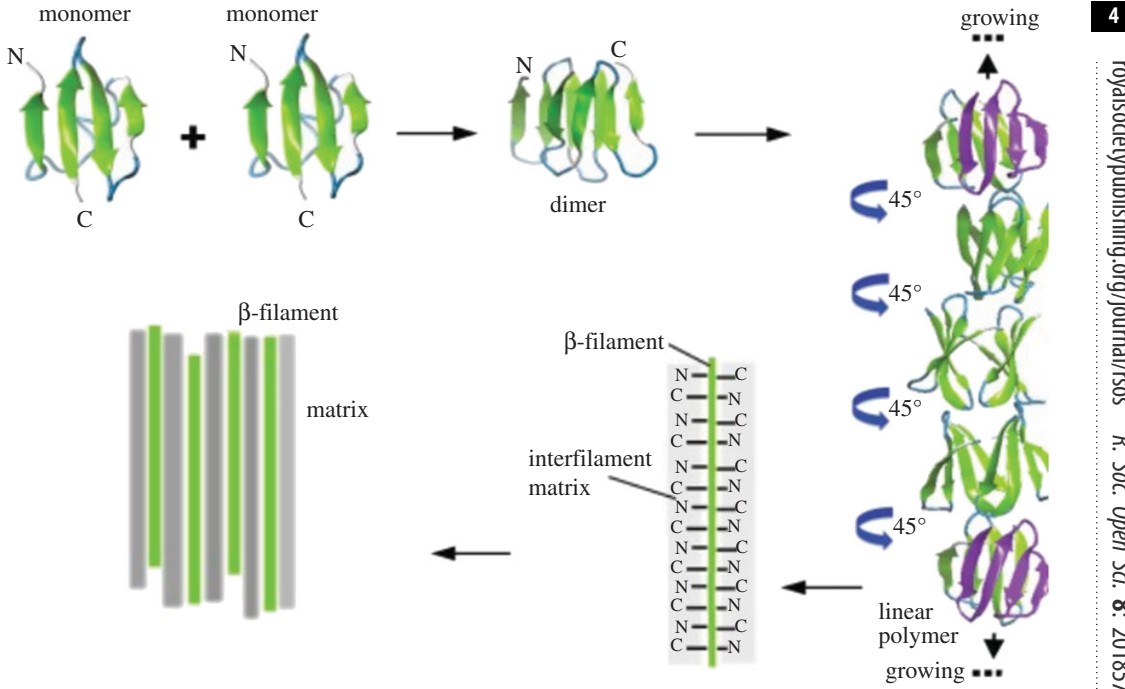

**Figure 1.** Schematic illustration of the arrangement of the CBPs into dimers and polymers, with the interfilament matrix formed by the N- and C-regions of the CBPs and cytokeratins (α-keratins). Reprinted from Alibardi [44]. Copyright (2016), with permission from Elsevier.

sauropsid-specific cytokeratins allows for the formation of disulfide bridges with cysteines in the N- and C-termini of CBPs, providing mechanical resistance [48] (figure 1).

At the time of this research, the search for Testudines proteins in UniprotKB (www.uniprot.org) yielded a little over 100 000 entries (January 2020). This number was reduced to 262 proteins when searching the order for keratins, while the same search on NCBI (www.ncbi.nlm.nih.gov/protein) gave 450 keratin entries for turtles (January 2020); the proteins were identified under a variety of names such as claw keratin-like, scale keratin-like or keratin-associated β-protein. While the protein name is an indication of a CBP, it has been found that these proteins can be expressed in multiple tissues [40], including the shell. Table 1 lists the species for which corneous β-proteins and α-keratins were identified in Uniprot, complemented by three additional species found in NCBI (full list given in electronic supplementary material, SM_Protein identification). Sequences for CBPs aligned using MUSCLE (Muscle 3.8.425 by Edgar [49]) show that over one-third of the CBP sequence in N-terminal (the first 70 residues that include the β-sheet region) is highly conserved, in particular the central region (figure 2 for *Chelonia mydas* and electronic supplementary material, figure S1 for *Pseudemys nelsoni*). The rest of the sequence is a high frequency of glycine-repeat regions G–X, G–G–X or G–G–G–X, where X is mostly tyrosine Y or leucine L (and occasionally phenylalanine F) residues. The amino acid composition of all CBPs known, of which only a few are from the green sea turtle, indicates that glycine is the predominant amino acid followed by high amounts of serine, tyrosine, proline, valine, leucine and cysteine (electronic supplementary material, figure S2) [50].

Due to the lack of complete genome sequences for marine turtles, proteomics identification is limited to the CBPs sequences of land turtles and a few sequences for the green turtle, making identification of unknown samples to any marine turtle impossible. Reference materials are, therefore, needed to build a database of the proteins targeted and determine the level of variation between species (robust markers need to be determined as degradation in ancient artefacts always leads to a loss of peptides that can make identification of a species more challenging). Here, proteomics analysis was performed on marine turtle scutes to identify species based on novel peptide sequencing of CBPs. Specimens were obtained from the Division of Amphibians and Reptiles of the Smithsonian's National Museum of Natural History (NMNH) for five species of sea turtles that grow scutes: green turtle, loggerhead turtle, hawksbill turtle, olive ridley turtle and Kemp's ridley turtle. The flatback turtle, geographically confined to Australia, was not included in this study as no samples were available at NMNH. The scute samples were processed for proteomics analysis by mass spectrometry and the sequences of

```
                                                                central
consensus                 MXXSSLCYPECGVARPSPVXGSXNEPCVRQCPDSEVVIRPSPVVVTJPGP    50
tr|M7BZP2|M7BZP2_CHEMY    MPFSSLCYPECGVARPSPVTGSCNEPCVRQCPDSEVVIRPSPVVVTLPGP    50
tr|M7C836|M7C836_CHEMY    MSCSSLCYPECGVARPSPVSGSFNEPCVRQCPDSEVVIRPSPVVVTIPGP    50

                          region
consensus                 ILSNFPQQSEVAAVGAPVVGAGXGGSFGLGGLYGYGGRYGGLYGLGXXGX   100
tr|M7BZP2|M7BZP2_CHEMY    ILSNFPQQSEVAAVGAPVVGAGLGGSFGLGGLYGYGGRYGGLYGLGRYGS   100
tr|M7C836|M7C836_CHEMY    ILSNFPQQSEVAAVGAPVVGAGYGGSFGLGGLYGYGGRYGGLYGLGGLG-    99

consensus                 XXXXXGYGGXXGXXGXCGYXGXYGYGGLXGXXXXXXXXXXXXXXXWXKXXX   150
tr|M7BZP2|M7BZP2_CHEMY    CGVLYGYGGLLGNGGYCGYPGLYGYGGLLGHGGYCGYPTQQEHPWKKEEP   150
tr|M7C836|M7C836_CHEMY    -----GYGGHYGYAGLCGYGGRYGYGGLSGF-------------WVKNKQ   131

consensus                 ZXXXXIXXXXXXXGXXXXXXXXXXXXXXXXXXXXLXLJXXXXX          194
tr|M7BZP2|M7BZP2_CHEMY    EMNSKIQIHPSSFGHGFQKCCANMAPPEGALCAQNLGLIAMLHF          194
tr|M7C836|M7C836_CHEMY    QIQREISEALCRHGSTW-------------IQRELCLLSAFVS          161
```

**Figure 2.** Alignment of M7BZP2 and M7C836 (Uniprot) from *C. mydas* using MUSCLE, with the central region highlighted. Amino acids are colour-coded based on polarity. Arginine and lysine residues are in bold. X indicates a residue where there is no consensus and J stands for the isomers leucine L and isoleucine I.

**Table 1.** Number of corneous β-proteins and α-keratin entries in turtles in UniprotKB and NCBI.

| Latin name | common name | β-protein-like entries | α-keratin entries | database |
|---|---|---|---|---|
| *Pelodiscus sinensis* | Chinese softshell turtle[a] | 37 | 24 | Uniprot |
| *Pseudemys nelsoni* | Florida red-bellied cooter | 18 | — | Uniprot |
| *Gopherus agassizii* | Agassiz's desert tortoise | 15 | 52 | Uniprot |
| *Apalone spinifera* | spiny softshell turtle[a] | 10 | 3 | Uniprot |
| *Chelonia mydas* | green turtle | 5 | 14 | Uniprot |
| *Platysternon megacephalum* | big-headed turtle | 5 | 28 | Uniprot |
| *Chrysemys picta bellii* | western painted turtle | 36 | 42 | NCBI |
| *Terrapene carolina triunguis* | three-toed box turtle | 2 | 36 | NCBI |
| *Gopherus evgoodei* | Sinaloan thornscrub tortoise | — | 37 | NCBI |

[a]Soft-shelled turtles.

the peptides determined by comparison with sequences from existing database complemented with bioinformatics de novo-generated sequencing. The novel peptide sequences were used to determine the species in seven comb fragments excavated in archaeological sites in France.

# 2. Material and methods

## 2.1. Reference samples

Scute samples were obtained from 17 specimens from five species at the Division of Amphibians and Reptiles, NMNH: table 2. Biological duplicates were processed and analysed separately and for each sample, analytical duplicates were carried out, resulting in a total of four runs for each specimen.

**Table 2.** Specimen of sea turtles sampled here with mass tested in extractions (1) and (2).

| species | catalogue number | sample name and type | provenance and collection date | (1) mg | (2) mg |
|---|---|---|---|---|---|
| *Chelonia mydas* green | USNM 132544 | CM132544 carapace | Republic of Marshall Islands, Bikar Island. Year = 1952 | 3.0 | 3.7 |
| | USNM 30756 | CM30756 carapace | Philippines, Bohol Island. Year = NA | 3.2 | 3.4 |
| | USNM 313726 | CM313726 carapace | Florida (USA), Indian River Lagoon system. Year = 1989 | 3.0 | 1.1 |
| | USNM 235887 | CM235887 carapace | Aldabra Islands, Grande Terre, Premiere L'Anse. Year = 1969 | 3.0 | 3.3 |
| *Caretta caretta* loggerhead | USNM 247944 | CC247944 carapace | Georgia (USA), Cumberland Island. Year = 1984 | 3.0 | 3.6 |
| | USNM 293677 | CC293677 carapace | China, Pingtan, Da Fu. Year = 1973 | 3.0 | 3.0 |
| | USNM 291936 | CC291936 carapace | Louisiana (USA), Rutherford Beach. Year = 1981 | 3.3 | 3.1 |
| | USNM 291938 | CC291938[a] carapace | Louisiana (USA), Rutherford Beach. Year = 1981 | 3.0 | 3.0 |
| | USNM 129257 | CC129257 carapace | Maryland (USA), Fairhaven Beach. Year = 1949 | 3.3 | 3.0 |
| *Lepidochelys olivacea* Pacific ridley | USNM 293679 | LO293679 carapace | China, Pingtan Island, Da Fu. Year = 1980 | 3.1 | 1.4 |
| | USNM 300004 | LO300004 carapace | Ecuador, Manta. Year = 1981 | 3.0 | 1.1 |
| | USNM 300009 | LO300009 carapace | Ecuador, Manta. Year = 1981 | 2.9 | 0.9 |
| *Lepidochelys kempii* Atlantic ridley | USNM 285140 | LK285140 carapace | New York (USA), Southampton Shores Beach. Year = 1987 | 3.2 | 1.4 |
| *Eretmochelys imbricata* hawksbill | USNM 117358 | EI117358c carapace | New Caledonia, La Foa. Year = NA | 2.9 | 2.4 |
| | | EI117358p plastron | | 3.0 | 1.9 |
| | USNM 220810 | EI220810 carapace | Philippines, Sulu Sea. Year = 1840 | 3.0 | 3.1 |
| | USNM 237700 | EI237700 carapace | Indonesia, Moluccas, Kampung Pasir Putih. Year = 1981 | 2.9 | 1.8 |
| | USNM 314860 | EI314860 carapace | Virgin Is (USA), Hans Lollik Island. Year = 1991 | 3.3 | 3.9 |

[a]The fragmented specimen was labelled as *L. Kempii*, but proteomics data showed it was *C. caretta*.

## 2.2. Comb samples

All archaeological samples were obtained from combs recently excavated in eighteenth- to nineteenth-century sites in France. Two combs were found in Nancy at the site of the parish cemetery of Trois-Maisons (Meurthe-et-Moselle) during excavations in 2010 [51]. The comb found in grave SP 97 dated to the last phase of use of the graveyard (1818–1842) has a single row of 14 teeth of which only seven

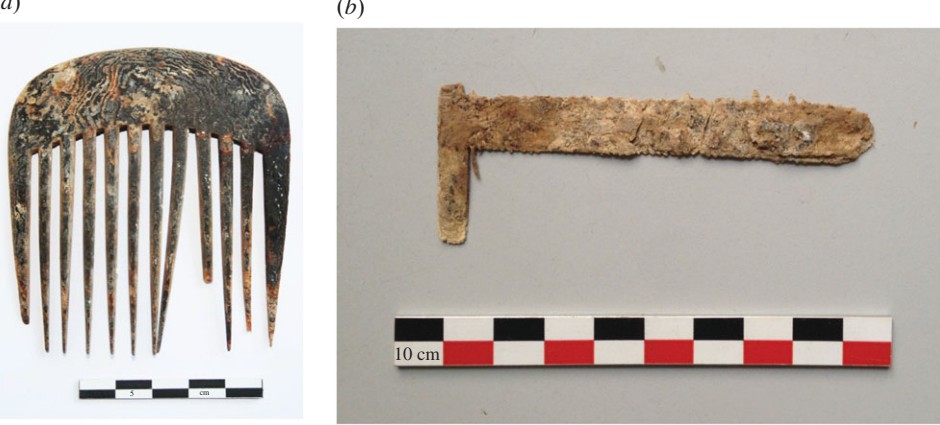

**Figure 3.** (*a*) SP 88, hairdressing comb and (*b*) Inv. 16914, grooming comb.

have been preserved. A sober decor consisting of a thin grooved applique adorns the upper part of the comb. By contrast, the second comb from the filling of the tomb SP 88 was complete (figure 3*a*). Both combs are hairdressing combs designed to maintain voluminous female hairstyles well known in the eighteenth century or to support a larger wig. A tortoiseshell comb was found in the tomb SP 1074 during the archaeological dig of the cemetery of the Protestant Hospital of La Rochelle (Charente), occupied between 1765 and 1792 [52]. Found under the skull of the deceased female (18–24 year old), this curved, 11.7 cm long comb has a single row of 14 teeth widely spaced, the majority of which are preserved. It fits perfectly with the standard of the second half of the eighteenth century which corresponds to the dating of the cemetery. The shape of the comb and its location behind the head of the deceased, in a functional position, suggests another hairdressing comb. Four combs and over 170 fragments of manufactured and raw tortoiseshell fragments were found during excavations of the Napoléon Court (1981–1986) on the site of the pyramid of the Louvre Museum in Paris where the workshop of the famous cabinetmaker of King Louis XIV, André-Charles Boulle burned in 1720 [11,53]. The comb analysed here (Inv. 16914, figure 3*b*) is a double-sided grooming comb, no teeth remain and its condition is poorly preserved. Two hairdressing combs come from excavations of the cemetery of Hôtel-Dieu in Lyon, graves SP 6536 and SP 6554, probably dated from the second half of the eighteenth century, and one last comb TAB-002 comes from the 2017 preventive excavations of courtyards in Rennes [54].

For each comb, one sample of 3–7 mg was available for protein extraction (electronic supplementary material, table S2); analytical duplicates were carried out, resulting in a total of two runs for each comb.

## 2.3. Sample preparation for proteomics analysis

A review of the CBP sequences available in database indicated that the proteins have few lysine K residues but enough arginine R residues to make them suitable for trypsin digestion, especially in N- and C-termini. In these sections of the proteins, the arginine residues are also well conserved (figure 1; electronic supplementary material, figure S1).

Reference scute samples and archaeological samples were cut in a dozen pieces with a razor blade. Proteins were extracted by overnight shaking in 200 µl of a solution of 8 M urea, 50 mM Tris and 50 mM tris(2-carboxyethyl)phosphine (TCEP) at pH 8.0. An aliquot of 100 µl supernatant was alkylated for 45 min in the dark with 11 µl of 400 mM of iodoacetamide for a final concentration of 40 mM. Samples were dialysed overnight in dialysis units at 3500 Da (Thermo Scientific™ Slide-A-Lyzer™ 3.5 K MWCO MINI Dialysis) with 100 mM Ambic (ammonium bicarbonate) at pH 8.0. The whole dialysed sample was then digested for 16 h with 0.5 µg of trypsin at 37°C. After digestion, the samples were acidified with 1% formic acid (FA) and the peptides extracted and purified by solid phase extraction with Empore SPE Extraction Disk (3M). The discs (Ø 0.1 cm) were washed with acetonitrile (1 min), conditioned with methanol (1 min) and washed with 0.1% FA solution (1 min) before being added to the samples and mixed for 3 h to allow peptide loading on the disc. Following a brief wash of the disc in 0.1% FA solution (1 min), the peptide mixtures were then eluted in 100 µl of 75 : 25 (v/v) acetonitrile : 0.1% FA for 1 h. All samples were then dried down on speedvac and resuspended in 10 µl of 0.1% FA.

## 2.4. Mass spectrometry analysis

The samples were diluted according to their starting mass and the injection volume adjusted to 0.5–1 µl. The peptides were first loaded onto an in-house packed Thermo BioBasic C18 precolumn (30 mm × 75 µm i.d.) after which they were separated on an in-house packed analytical column (210 mm × 75 µm i.d.) made of the same stationary phase, using a Thermo Scientific Dionex Ultimate 3000 UHPLC system with the following gradient: 2% B 0–8 min, 55% B 98 min, 90% B 100–103 min, 2% B 104–120 min, where buffer A is 0.1% FA in $H_2O$ and buffer B is 0.1% FA in ACN. The UHPLC was directly coupled to a Thermo Scientific LTQ Velos Dual Pressure Linear Ion Trap mass spectrometer which analysed the peptides in positive mode using the following parameters: MS1 60 000 resolution, 100 ms acquisition time, $1 \times 106$ automatic gain control (AGC), MS2 15 000 resolution, 250 ms acquisition time, $5 \times 105$ AGC, top 8, 30 normalized collision energy (NCE) higher-energy collisional dissociation (HCD).

PEAKS 8.5 (Bioinformatics Solutions Inc.) was used to search the RAW data for matches against publicly available sequences. Keratin-named sequences for the order Testudine (or Chelonians) were imported on 23 January 2020 from Uniprot and NCBI, and combined into a database (Uniprot + NCBI keratin Testudines). Because of the limited number of sequences, PEAKS was used in SPIDER mode to search for amino acid substitutions and DENOVO mode to identify new sequences. For each specimen and archaeological samples, all runs were pooled into one, resulting in one output file for each sample. Searches were carried out using trypsin as enzyme with semispecific digest mode, one missed cleavage, peptide mass tolerance (PMS) of 10 ppm, fragment mass error tolerance (MS/MS) of 0.02 Da, carbamidomethylation as a fixed modification, and deamidated (NQ) and oxidation (M) as variable modifications, and a maximum variable PTMs per peptide set at three. Results were filtered using a peptide score of $-10\lg P \geq 25$ for peptide spectrum matches, a protein score of $-10\lg P \geq 20$, and one unique peptide, and a cut-off of 90% for de novo peptides.

Identified de novo peptides were added to the in-house database (Uniprot + NCBI + Denovo keratin Testudines) and a final search conducted in PEAKs with the parameters set above, adding oxidation (PYHW), dihydroxylation (YW) and pyroglutamate (Q) as variable modifications. PEAKs PTM was run with a maximum variable PTMs per peptide set at three. Results were filtered using a peptide false discovery rate (FDR) of 1% for peptide spectrum matches, a protein score of $-10\lg P \geq 20$, and 0 unique peptide.

## 2.5. PCA classification of modern and archaeological samples

Each modern sample was classified using principal component analysis (PCA) using the highest $-10\lg P$ PEAKS score of each peptide. Unique peptides were selected using Excel 365 and then PCA was performed in R (v. 3.6.1) with prcomp with no scaling. The archaeological samples (highest $-10\lg P$ score) were subsequently classified using the modern sample classification system. Clustering was performed using the kohonen self-organizing map (SOM) method with a 4 × 4 square grid and 10 000 replicates. Elbow $k$-means clustering was used to form clusters from the SOM.

# 3. Results

## 3.1. Peptide classification

Based on known CBPs, a classification of the most frequently encountered sequences in the sea turtles was attempted, and the protein sequences divided into peptides A to F (table 3); however, the list is not exhaustive. Figure 4 and electronic supplementary material, figure S3 show examples of known CBPs with position of classified peptides.

The conserved portion of the CBPs is divided into peptides A to D. Peptides A, B, C and F (C-terminal) are fixed in length due to the presence of arginine residues and, for A and F, due to being termini peptides. Most of peptide D belongs to the portion of the sequence containing the β-sheet; for lack of lysine or arginine in this long part of the CBPs, it is arbitrarily stopped before the glycine-rich region. Results on the sea turtles show an abundance of peptides identified at non-tryptic cleavages in D as well as overlapping with peptides C and E1. Because of the many random cleavages and the highly conserved nature of this section, D was not used for species differentiation. The glycine-rich region was divided into four peptidic segments E1–E4. Peptides in this section vary in length (with

(a)    tr|M7C836|M7C836_CHEMY claw keratin OS = *Chelonia mydas*

**A**        **B**        **C**            **D**
MSCSSLCYPECGVAR PSPVSGSFNEPCVR QCPDSEVVIR PSPVVVTIPGPILSNFPQQSEVAAVGAPVV *69*
   **E1**                    **E2ₐ**                      **E3**
GAGYGGSFGLGGLYGYGGR YGGLYGLGGLGGYGGHYGYAGLCGYGGR YGYGGLSGFWVK *128*

NKQQIQREISEALCRHGSTWIQRELCLLSAFVS *161*

(b)    tr|B7FCD1|B7FCD1_9SAUR beta-keratin-like protein OS = *Pseudemys nelsoni*

**A**        **B**        **C**            **D**
MSCSSLCYPECGVAR PSPVSGSCNEPCVR QCPDSEVIIR PSPVVVTIPGPILSNFPQQSEVGAVGAPVV *69*
                  **E1**                        **E2_b**
GAGYGGSFGLGGLYGYGGH YGGLYGYGGLGGYGGR YGYGGGYGGGYGGLCGYGGR *124*
   **E3**                **E4ₐ**                          **F**
YGYGGLSGYGGR YGGLCGYGGGYGGGYGYGGACGSGVSCHR YLSGSCTPC *174*

**Figure 4.** Examples of proteins with location of peptides A–F, as described in table 3: (a) green turtle (161 residues long) and (b) land turtle Florida red-bellied cooter (174 residues long).

**Table 3.** Classification of CBP peptides most commonly identified in the sea turtles, from A to F. Based on comparisons with known CBPs, the type of tryptic cleavage in N- and C-terminus is best estimated. The template sequence was determined from all identified and validated sequences for any given peptide, as reported in electronic supplementary material, with X indicating variable residues.

| name | Tryptic cleavage N/C-termini | length of residues | species | template sequence |
|---|---|---|---|---|
| A | N and C | 14 | all | XXSXLXYPEXGVAR |
| B | N and C | 14 | all | PXPXXGSXNXPXVR |
| C | N and C | 10 | all | QCPDSEVVIX |
| D | N | 30 | all | PSPVVVTIPGPILSNFPQQSEVAAVGAPVV[a] |
| E1 | C | 19 | all | GXGXGGSFGXGGLYGYGGR |
| E2ₐ | N and C | 28 | *Chelonia mydas* | YGGLYGLGGLGGYGGHYGYAGLXGYGGR |
| E2_b | C | 16 | all | YGGLYGYGGXGGYGGR |
| E2_c | N and C | 9 | all | LGGFGGLXR |
| E2_d | C | 9 | all | XGXXXGYGR |
| E3 | N and C | 12 | all | YGXGGLXGYGGR |
| E4ₐ | C | 25 | all but *Chelonia mydas* | YGGXCGYGGGYGYGGLSGSGVSXHR |
| E4ₐ | C | 28 | all | XGYGGLGGYXGGYGYGXGLXGSGVSXHR |
| E4_b | N and C | 8 | all | YGGXXGYR |
| F | N | 9 | all | YLXGXCGPC |

[a]A few variations were identified with SPIDER in this sequence (data not shown) at positions I12, N15, Q18, E21, A27, V30.

additional glycine and tyrosine-rich segments), and some had to be further subcategorized, i.e. peptides E2 and E4. The classification was based on alignments and checked manually, but there are instances where short peptides could fit in different locations (for example, E3 in E2_b).

## 3.2. De novo sequencing

The initial search on Uniprot + NCBI sequences yielded a large number of peptide matches, the majority from the PEAKs DENOVO output. De novo sequences identified with an average local confidence (ALC) greater than or equal to 90% were selected for further evaluation. For each peptide, sequences that were

most frequently recurring were concatenated into one fasta sequence, resulting in fasta sequences for peptides A, B, C, etc. The sequences were added to the in-house database (Uniprot + NCBI + Denovo keratin Testudines) and all files searched again. After filtering peptides at an FDR of 1% and manual validation of the peptides based on complete or near-complete y ions series, a total of 187 marker sequences were accepted (electronic supplementary material, tables S1 and SM_Peptide details) of which only 14% were obtained from known turtle CBPs; the majority of sequences were thus determined from de novo sequencing. The peptides for which the most combinations were found were peptides A, B and E4a with, respectively, 34, 53 and 47 marker sequences.

For marker validation in each species, outliers were removed by setting a score cut-off for each individual sequence. The score cut-off was determined by calculating the score's median absolute deviation (MAD) from all peptide-to-spectrum matches (PSMs) with carbamidomethylation of cysteines or unmodified, and subtracting the MAD from the score's median value. To visualize the distribution of markers in all samples, Peaks output files were imported as .mzid files into the program PACom—Proteomics Assay COMparator (v. 1.87) [55]. Heatmaps based on the number of PSMs were created for each marker by filtering scores above the calculated cut-off; heatmaps and mass spectra are shown in electronic supplementary material, SM_Peptide details for each marker sequence. Validated markers for each species are given in electronic supplementary material, tables S1.

## 3.3. Species markers

For each characterized peptide, template sequences in table 3 indicate the residues X where substitutions occur in the validated markers; the other residues are largely or always preserved among markers. In peptide B, for instance, $PX_2PX_4X_5GSX_8NX_{10}PX_{12}VR$, all the residues marked in bold remain unchanged across species (only in two markers—B-30 and B-34—the VR tag at the end is replaced by PR), making the pattern of the sequence highly predictable. Markers unique to species usually have exclusive substitutions: for example A-15 with S3 → E3 is unique to *E. imbricata*. In peptide B, markers B-1 to B-3.2 characterized by a leucine instead of a valine in $X_4$ are unique to *C. mydas*; similarly, all markers with a leucine in $X_{10}$ are unique to *C. caretta*. Three markers C (C-5, C-6 and C-7) with arginine substituted by lysine are found uniquely in *C. mydas*.

Peptide $E4_a$ represents perhaps the most interesting segment of the CBPs. The peptide is characterized by a conserved -GSGVS**X**HR tag on C-terminus where **X** is either alanine A or cysteine C. The peptide is 25–28 residues in length with markers $E4_a$-1 to -9 characterized by the 25-residues sequence - -YGGXCGYGGGYGY-GGLSGSGVSXHR (one subtracted residue in the middle), and markers $E4_a$-10 to -28 XGYGGLGGYXGGYGYGXGLXGSGVSXHR with 28 residues; all the identified sequences come from the PEAKs DENOVO mode, and were manually validated.

The shorter markers $E4_a$-1 to -9 are not found in *C. mydas*, but are otherwise well conserved across all other species. Only one marker (E4a-7) with a unique substitution $G_{14} \rightarrow A_{14}$ is unique to *C. caretta*.

Markers $E4_a$-10 to -28 are usually found in multiple forms in each species with residue $X_1$ being alternatively alanine A, cysteine C, proline P or tyrosine Y. The markers with Y in $X_1$ are found uniquely in *E. imbricata* ($E4_a$-24.2, 25.2, 26.3 and 27.3). Other substitutions unique to some species are: L6 → F6 in markers $E4_a$-10 to -12 is unique to *L. olivacea*, G8 → A8 in $E4_a$-13 is unique to *C. mydas*, L19 → V19 in markers $E4_a$-23.1 and 28 is unique to *L. olivacea,* and Y9 → S9 in markers $E4_a$-10, 11, 12, 14 and 15 is unique to *L. olivacea*. Notably, few E4a markers were found in *L. kempii*, so the differentiation of the species based on this marker is inconclusive until more specimen of *L. kempii* can be tested. There is indeed a lower count of peptides for *L. kempii*, represented by only one specimen after a second specimen collected as *L. kempii* was clearly matched with *C. caretta* based on the present analysis. The specimen USNM 291938 was partial and had been collected together with USNM 291936 at the same location and time. Either the scutes had been misidentified or the fragments mixed up with the *C. caretta* specimen. In spite of the lower count of *L. kempii* peptides, two markers were found uniquely in the species, both from E4b.

## 3.4. Post-translational modifications

Evaluation of post-translational modifications with PEAKs PTM indicates some phosphorylation (+79.97 Da), for the most part in peptides A, B and E4a and on serine residues. Many of the phosphorylated serine residues were followed by an alanine (phosphorylated sites indicated in SM-Phosphorylation for peptides A, B and E4a); for instance, in E4a, most serine in the SAHR pattern

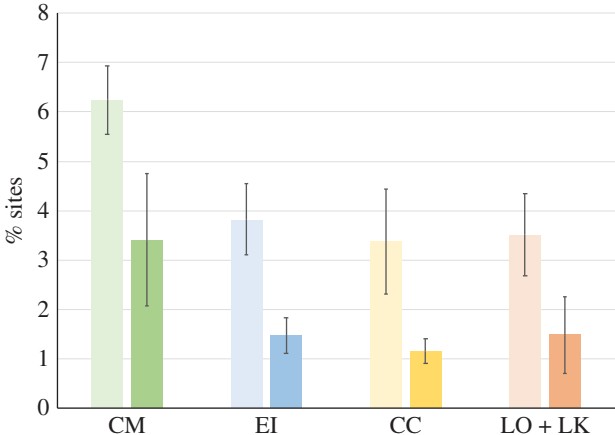

**Figure 5.** Percentage of sites with oxidation (light colour) and dioxidation (darker colour) for markers A, B and E4a combined, with standard deviation. CM, *C. mydas*, $n = 4$ specimens; EI, *E. imbricata*, $n = 5$; CC, *C. caretta*, $n = 5$; LO, *L. olivacea* and LK, *L. kempii*, $n = 4$.

were phosphorylated but none of the serine in the SCHR pattern. Deamidation was negligible due to the low frequency of asparagine and glutamine residues.

By far the most common modifications were oxidation (+15.99 Da) and dioxidation (+31.99 Da) on the proline, tyrosine, histidine and tryptophan residues. Figure 5 shows the percentage of sites with oxidation and dioxidation for peptides A, B and E4a combined as these peptides again bear the large majority of the oxidative modifications. For each marker, the number of potential sites was determined (number of P, Y, H and W) and multiplied by the number of PSMs detected for that marker, thus indicating the total number of potential sites for oxidation. The percentage of oxidation and dioxidation sites was calculated from the number of sites detected with oxidation and dioxidation (obtained in PACOm). For each genus (*L. olivacea* and *L. kempii* were combined into the *Lepidochelys* genus), the percentage of oxidation was about double the percentage of dioxidation. Percentages of oxidation and dioxidation were similar for *E. imbricata*, *C. caretta* and the *Lepidochelys* sp. *Chelonia mydas*, however, shows levels of oxidation and dioxidation about twice as high as the other species.

## 3.5. Archaeological combs

The *k*-means SOM clusters were found to represent the following species: (i) *C. caretta*, (ii) *C. mydas*/ *L. kempii*, (iii) *L. olivacea*, and (iv) *E. imbricata*. Based on these clusters, the combs were identified as deriving from *E. imbricata* (SP1074, SP6554, SP88, SP97), *L. olivacea* (Inv16914) or *C. mydas*/*L. kempii* (SP6536, TAB002) (figure 6a).

Unlike the PCA analysis, the archaeological samples were all identified as *E. imbricata* by biomarker analysis. Samples with the best matches to *E. imbricata* were SP 88 and SP 97 from Nancy, SP 1074 from La Rochelle and SP 6554 from Lyon with between 69 and 75% of *E. imbricata* markers identified. The samples with lower peptide counts, SP 6536 from Lyon, Inv. 16 914 from Paris and TAB-002 from Rennes, were matched with between 52 and 56% of markers. The clustering of the two groups on figure 6b reflects the difference of markers identification, probably due to preservation of the sample. Figure 7 shows the markers unique to *E. imbricata* and their identification in the combs: of the 20 markers, 16 were found in one or more combs and four of these markers (C2, C8, E2c-4 and E4a-19.1) were found in all combs. Heatmaps matching all markers from *E. imbricata* are shown in electronic supplementary material, figure S4.

Hydrolytic damage is common in ancient proteins and leads to loss of peptides and random cleavages. Here, markers E4 are the most affected in the combs (for example, only two E4 markers were detected in SP 6536, see electronic supplementary material, figure S4), followed by markers A (figure 8a). In addition, the percentage of PSM with undetermined cleavages was calculated for all peptides combined (for each individual peptide, see electronic supplementary material, figure S5). That percentage is on average 30% in *E. imbricata* (figure 8b) and nearly doubles in the archaeological combs. In addition, very few peptides belonging to α-keratins were found in the archaeological samples compared with the references, indicating rapid loss of these proteins (data not shown).

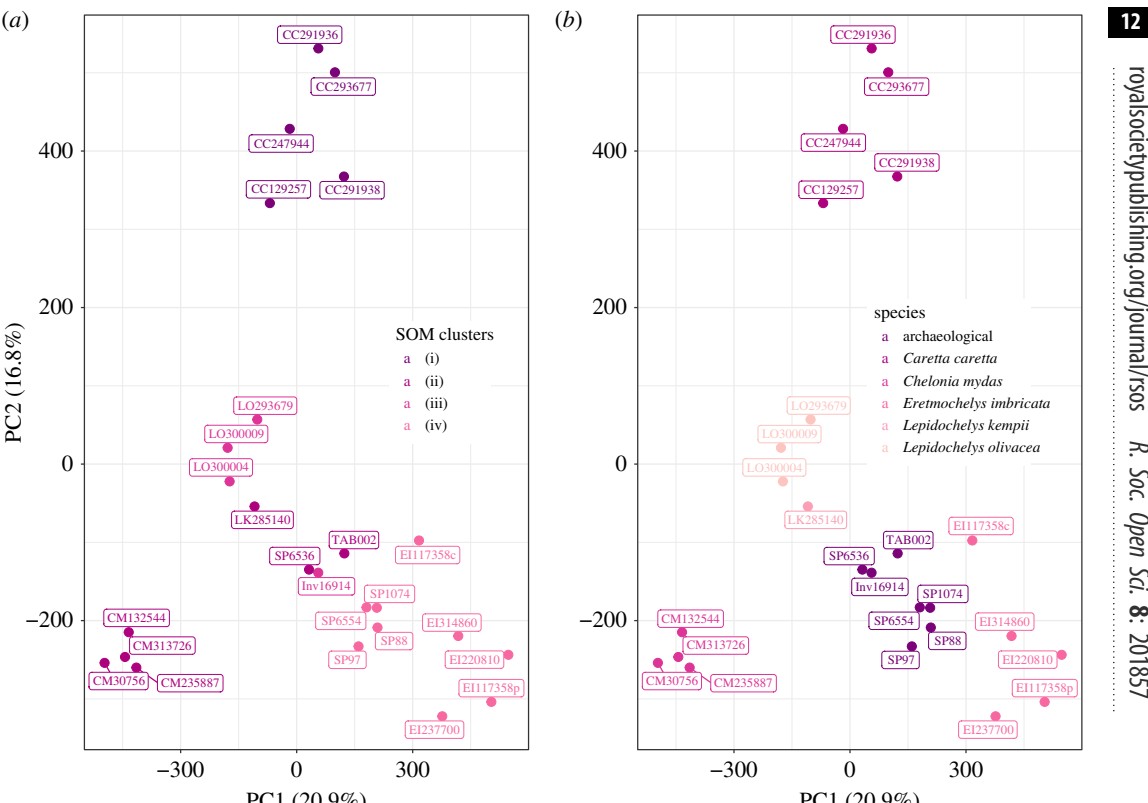

**Figure 6.** PCA analysis of each sample with (*a*) SOM clustering or (*b*) species/archaeological type. The *k*-means SOM clusters represent the following species: (i) *C. caretta*, (ii) *C. mydas/L. kempii*, (iii) *L. olivacea* and (iv) *E. imbricata*.

## 4. Discussion

### 4.1. Observations on the primary and secondary structures of sea turtle's CBPs

The data obtained on the sea turtles indicate that the first half of the sequence has retained the conserved pattern seen in certain land turtles such as *P. nelsoni*, *P. sinensis* or *C. picta bellii* with which the sea turtles were best matched in the initial database search (see electronic supplementary material, SM_Protein identification). Indeed, the frequency of tryptic markers found for peptides A, B and C indicates the position of the arginine residues must have remained largely conserved in the sea turtles. Furthermore, the number of markers discovered for peptides A and B is consistent with the high number of CBPs in land turtles. In *P. sinensis* and *C. picta bellii*, 37 and 36 CBP entries are recorded, while for *C. mydas*, 31 peptide A and 36 peptide B markers were identified. Peptides C, D (data not shown) and E1 indicate a more homogeneous region with less than 10 markers identified in each peptide. However, that region is not easily characterized by trypsin digestion due to the absence of arginine and lysine. Some CBPs in land turtles indicate that the arginine in peptide C is substituted by residues other than lysine (electronic supplementary material, figure S1), suggesting more sequences could be found. The absence of arginine and lysine between peptides D and E1 is also responsible for the occurrence of random cleavages and some very long fragments (data not shown). In that regard, E1 (that has a high percentage of PSM with undetermined cleavages, see electronic supplementary material, figure S5-E1) might not be considered a reliable marker for species identification.

The C-terminal side of the CBPs is equally difficult to fully characterize as it presents a large disparity in the length of the segments and positions of the arginine residues. Thus, the section covered by E2 and E3 had to be subdivided into several peptides, and each species has a dozen or more markers in that region. There is, however, a high number of markers for peptide E4a for *E. imbricata* (21), *C. caretta* (22) and *L. olivacea* (27), underlining a sequence pattern commonly seen in land turtles too but with different lengths. Finally, in C-terminal is peptide F, commonly found in

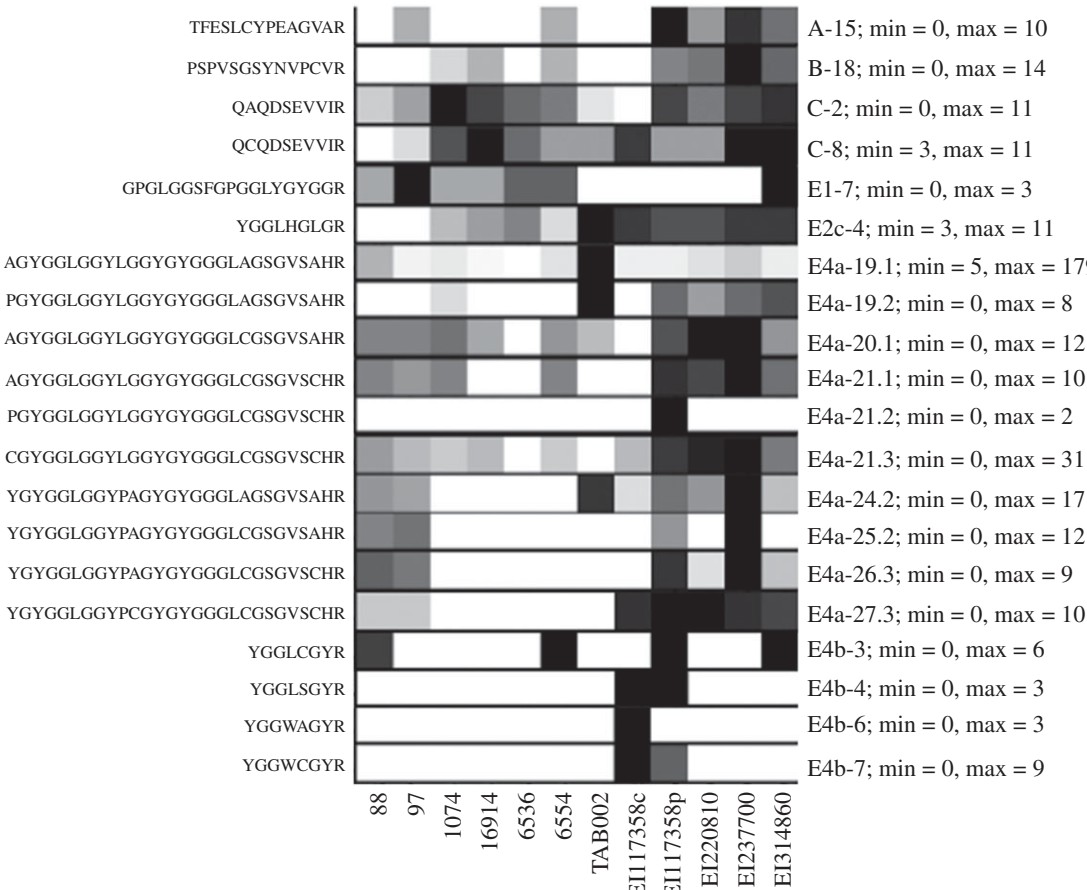

**Figure 7.** Heatmaps of the individual markers identified uniquely in *E. imbricata* (generated in PACom with calculated cut-off score applied to each marker); min and max indicate the minimum (white) and maximum (black) cell values (in number of PSMs).

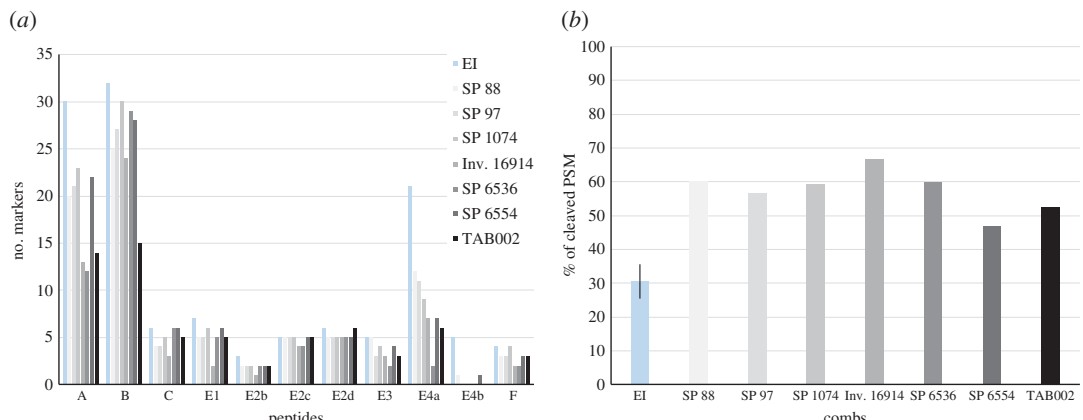

**Figure 8.** (*a*) Number of markers identified in the *E. imbricata* species and in the archaeological combs; (*b*) percentage of PSM with undetermined cleavages to the total number of PSM for all peptides. EI is given as an average with standard deviation of all five specimen of *E. imbricata*. Combs are indicated by their accession numbers: SP 88, SP 97, SP 1074, Inv. 16914, SP 6536, SP 6554 and TAB002.

CBPs of *P. nelsoni*, *P. sinensis* or *C. picta bellii* (figure 3*b*); only four markers were identified and they were found in all genera, indicating a very conserved C-terminal segment in all sea turtles.

The high cysteine content of the N- and C-termini is demonstrated by the abundance of cysteine residues found in peptides A, B, C and in E4 and F. Furthermore, peptides A and F have markers found in at least two species (only marker A-15 is unique to *E. imbricata*), suggesting the termini parts of the CBPs have diverged the least between the sea turtles species. The formation of the disulfide bridges between the CBPs and the cytokeratins, as mentioned earlier [48], and the higher conservation

of the peptides across species might be explained by the important configuration role, similar to the conservation of the central region.

Based on studies of the β-pleated region [43,56], the turtles' central region corresponds to the following sequence (in bold, from M7BZ2_CHEMY), centred around peptide C and most of D, and containing four β-sheets (highlighted in yellow):

PSPVTGSCNEP$_{25}$**CVRQ**CP$_{31}$DSE**VVIR**P$_{39}$SP$_{41}$**VVVT**LP$_{47}$GP$_{49}$**ILSN**FP$_{55}$**QQSEV**AAVGAPVV

The first β-sheet would begin at P25 (last proline in peptide B); notably, all the proline residues in peptides B and D are conserved for all species. In peptide C, sequences with a proline in P31 are found in all species, but in two sequences (C-2 = QAQDSEVVIR and C-8 = QCQDSEVVIR), P31 is substituted by a glutamine Q31. Interestingly, these two peptides are only identified in *E. imbricata*; the substitution in this crucial part of the sequence for the three-dimensional configuration of the CBPs might play a role in the flexibility and the superiority of this species' shell for tortoiseshell.

Finally, peptide E2$_a$ is unique to *C. mydas*, for which two sequences were identified and vary by just one amino acid residue. It is characterized by a histidine in position 104 (M7C836_CHEMY, figure 3*a*); in peptide E2$_b$, mainly found in the other species, this histidine is substituted by an arginine.

## 4.2. Species identification from CBPs

Less than one-fifth of markers (33) are found in all five species, but this increases to 30% (56 markers) when considering all four genera. *Eretmochelys imbricata*, *C. caretta* and the *Lepidochelys* sp. have 21 markers in common, while *C. mydas*, *C. caretta* and the *Lepidochelys* sp. have seven markers in common. This is consistent with a recent study of the phylogenetic analysis of collagen sequences placing *C. mydas* in a separate group than the four other species [22]. Based on the novel CBPs data, PCA analysis shows a clear separation of species in four clusters, corresponding to their genus. Further separation, based on sample origin was not apparent at this stage. *Eretmochelys imbricata* had the highest number of specific markers with 20 markers, followed closely by *C. mydas* (19), *C. caretta* (17) and *L. olivacea* (15). Markers specific to *C. mydas* were found in peptides B, C, E2a and E4a, while markers specific to the other species were found mainly in E4a and b, as well as C for *E. imbricata*, B for *C. caretta*, and E2b for *Lepidochelys* sp.

The archaeological samples were all unambiguously identified as hawksbill turtle, with between 64 and 93 of a total of 124 markers present in *E. imbricata*. Furthermore, all archaeological samples had at least four markers unique to *E. imbricata*: QAQDSEVVIR (C-2), QCQDSEVVIR (C-8), YGGLHGLGR (E2c-4) and AGYGGLGGYLGGYGYGGGLAGSGVSAHR (E4a-19.1). One unique marker, E1–7 GPGLGGSFGPGGLYGYGGR, was found in all combs but TAB-002 (figure 7), and was validated in specimen EI314860 only, the sole Caribbean Islands specimen tested here (see electronic supplementary material, SM_Peptide details). This finding offers interesting prospects for a more localized identification of the samples. Indeed, the archaeological combs are believed to be made from tortoiseshell harvested from the Caribbean as part of the trans-Atlantic trade between the Caribbean islands and France. However, these results have been taken with caution as de novo data suggest that this marker might be present in other specimens with missed cleavages. As mentioned above, E1 might be an unreliable marker for species identification. Further research will be necessary to determine if specimens from different stocks show differences in the expression of their CBPs, as some molecular studies have shown for the Atlantic and Indo-Pacific stocks of *E. imbricata* [38].

Focused here on the exhaustive search for the species to which the hard animal material used to shape the finished product belonged, these analyses made it possible to identify that the combs had been made from turtle scales, and that they all belong to the same species, *E. imbricata*. In addition, it would be relevant to carry out archaeological analyses on the 175 raw and ornamental fragments discovered during archaeological excavations of the Grand Louvre and linked to the rejects from the Boulle workshop [53]. A series of analyses could also be carried out on certain Boulle-style furniture being restored. The combined results of these analyses would make it possible to continue investigations into the origin of these sea turtle scales and to better understand the supply circuits of the artist and his workshop, in particular his influence and perhaps even his demand on the choice of goods. The constitution of the international database will also provide some answers to help better understand this particular material.

## 4.3. Tortoiseshell degradation

Tortoiseshell grows in layers, accumulating in the scutes; O'Connor *et al*. [12] have described the degradation of tortoiseshell as proceeding from the edges of an object along the layers that separate,

degrade into a granular state and eventually disintegrate. The location of the sampling is, therefore, critical to obtain good proteomics results. For instance, EI117358c is closer to the cluster of less-preserved archaeological combs than the other four *E. imbricata* samples. The sample, taken from an exposed area on the edge of the carapace, turned out to be of lower quality than EI117358p taken from the plastron. Similarly, samples first taken from flaky and brittle areas of SP 6536 and SP 6554 gave results of lower quality (data not shown) than the samples presented here, taken from the hardest core.

Higher degradation observed in peptides A and E4 seems to indicate preferential degradation at the N- and C-termini, and further away from the β-sheet region whose structural arrangement might enhance preservation in the central region and surrounding areas. The N- and C-termini are not part of the β-filament, but are parts of the matrix (figure 1). In the same way that the amorphous keratin-associated proteins (KAPs) are degraded more readily than the IFPs in archaeological wool [57], the results here indicate that the CBP's peptidic segments making up the matrix are also the first to degrade. Furthermore, the disruption of the regions that form disulfide bridges with the α-keratins (themselves rapidly lost) is consistent with the observed separation of the layers in archaeological tortoiseshell.

# 5. Conclusion

This study describes for the first time the proteomics analysis of turtles scutes from marine turtles with the objective of differentiating potential sources of tortoiseshell. It focuses on the corneous β-proteins, as they are the predominant protein family in turtle scutes. Preliminary examination of the α-keratins pointed to more conserved sequences across all species, with few species markers identified (data not shown). Furthermore, these proteins appeared more degraded in the archaeological combs, making them of little use for species identification.

Although challenging due to the nature and large number of corneous β-protein sequences, a series of distinct peptides were characterized, trying to follow as much as possible tryptic cleavages as known from CBPs from land turtles. The traditional protocol employed here with trypsin as proteolytic enzyme was efficient to characterize the N-terminus of the sequence (peptides A, B and C) and C-terminus (peptides E4 and F). Some sequence patterns in the glycine-rich region also emerged (peptides E2 and E3), although more might be found due to the changing position of the tryptic sites. For the core of the sequence, better strategies need to be developed due to the long portion without suitable tryptic sites. However, as the core of the sequence is well conserved between species, it presents less interest for species identification and thus was not used for that purpose. Characterization of new peptides relied heavily on assisted de novo sequencing, using the de novo function of PEAKs. The high quality of the MS/MS spectra (with near-complete y-ion series) allowed validation of 187 markers across five species of marine turtles with a high number of unique markers found in all species. Only *L. kempii* was limited to two unique markers, most likely due to the poorer representation of the species in the dataset (one specimen instead of two as the second specimen sampled was identified by proteomics as *C. caretta*).

A large number of markers unique to species were found, in particular in peptides B and E4. Of the 20 markers unique to *E. imbricata*, 12 were found in one or more archaeological samples and four were found in all samples, giving high confidence in the identification of even the most degraded combs. Interestingly, two of these markers, QAQDSEVVIR (C-2) and QCQDSEVVIR (C-8), have a unique substitution at the third residue where proline is replaced by a glutamine. As this proline has an important role in the formation of the β-pleated sheet, this substitution in the hawksbill turtle might be critical to the malleability of the turtle scute from this species, making it more amenable for tortoiseshell.

As species get sequenced in coming years, it will add confidence in the identification of the markers established from de novo sequencing and resolve issues of sequence homology. In the meantime, this study has shown that automated and targeted de novo sequencing could be a powerful tool to characterize unknown proteins in species without a sequenced genome, and a faster approach than DNA sequencing. Furthermore, the unexpected high number of markers revealed a high level of variability, possibly even within species as not all markers were equally distributed in all specimens. Future research might indicate whether these rarer species markers can point to individuals or groups of individuals. Proteomics genotyping has indeed shown promise for the forensics analysis of hair from single individuals based on genetically variant peptides in hair keratins [58,59]. Such an approach on marine turtles would be a valuable tool against illegal trade of tortoiseshell.

Data accessibility. The mass spectrometry data for LC-MS/MS have been deposited on MassIVE with the dataset identifier MSV000086219 (ftp://massive.ucsd.edu/MSV000086219/).

Authors' contributions. J.S. and C.S. initiated the study. C.S. conducted the proteomics analysis and analysed data. T.C. performed the PCA analysis. C.S. interpreted data and wrote the paper, with contributions from J.S. and T.C. All authors reviewed and approved the manuscript.

Competing interests. We declare we have no competing interests.

Funding. The proteomics analyses were carried out at MCI's Proteomics and Biomolecular Mass Spectrometry Laboratory, supported by MCI's Federal and Trust Funds and the Andrew W. Mellon Foundation Directorship Endowment (grant no. G-40700657).

Acknowledgements. The 'Transatlantic Connections' research programme is supported by the LandArc Laboratory in France, the Archeometry Research Group of Laval University in Quebec (GRAUL) and the Center Michel de Boüard of the University of Caen Normandie, France (CRAHAM UMR 6273, CNRS). For providing the samples, we thank Addison Wynn at the Smithsonian's National Museum of Natural History; Amélie Berthon, Bureau d'étude Eveha (Rennes sample); Cécile Pillard-Jude and Myriam Dohr, Inrap (Nancy samples); Néguine Mathieu and Anne-Laure Goisnard, département Histoire du Louvre, Musée du Louvre, Paris (Louvre sample); Jean-Paul Nibodeau, Inrap (La Rochelle sample); Stéphane Ardouin and Viannay Rassart, Service archéologique de la Ville de Lyon (Lyon samples). The proteomics analyses were carried at MCI's Proteomics and Molecular Mass Spectrometry Laboratory and supported by MCI's Federal and Trust Funds and the Andrew W. Mellon Foundation—Directorship Endowment.

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
