## [Peer Review File · Royal Society Open Science]

Review History

RSOS-201857.R0 (Original submission)

Review form: Reviewer 1

Is the manuscript scientifically sound in its present form?

Yes

Are the interpretations and conclusions justified by the results?

Yes

Is the language acceptable?

Yes

Do you have any ethical concerns with this paper?

No

Have you any concerns about statistical analyses in this paper?

No

Recommendation?

Major revision is needed (please make suggestions in comments)

Comments to the Author(s)

This manuscript reports the proteomic analysis of shell samples of turtles and tortoiseshell samples from archeological finds (combs). The authors present peptide sequence data that can be used for the identification of the species from which tortoiseshell was prepared. The data are sound and the manuscript is written well. The results are interesting for specialists in this field. A wider readership will be interested in the general approach of using proteome analysis for research questions outside of classical biology.

Major comments:

This study makes the best use of the available sequence data and generally gives convincing results. However, it is important to emphasize that the identification of species using proteomics is currently limited in power by the lack of the complete genome sequence of *Eretmochelys imbricata* and many other species of turtles. The implications of this problem should be discussed. The authors should also discuss potential improvements that can be expected when complete genome sequences of more turtle species become available, as promised by the Vertebrate Genomes Project.

Figure 6: The legend and the description of this figure in the text are not sufficient to explain the content and to allow interpretation.

Beta-keratins/CBPs are the main proteins of the tortoise shell but there are also other proteins which may be useful for the identification of species. Please discuss this option and state whether or not you have tested or will test this approach.

Minor comments:

Table 3: The tryptic cleavage should have occurred only at the N-terminus of peptide F, not at "N and C". Please check.

Figure 2: Please explain the meaning of letter J in the consensus sequence.

The letter ß does not mean beta, but a special form of "s" in German language. Please change to the real beta symbol β throughout the text.

Page 20 (numbering of the merged pdf file), line 38: Please check if the journal allows highlighting by color in the main text.

Review form: Reviewer 2

Is the manuscript scientifically sound in its present form?

Yes

Are the interpretations and conclusions justified by the results?

Yes

Is the language acceptable?

Yes

Do you have any ethical concerns with this paper?

No

Have you any concerns about statistical analyses in this paper?

No

Recommendation?

Accept with minor revision (please list in comments)

Comments to the Author(s)

Review of Solazzo et al. 2020:

“Creation of a peptide database of corneous beta-proteins of marine turtles for the identification of tortoiseshell; Archaeological combs as case study”

General comments

Overall, this is a well written, concise manuscript outlining the proteomic analysis of tortoiseshells, and the application to archaeological identification. I have some general comments that I will briefly touch on here, with more specific comments for each section detailed below. My recommendation is for acceptance with minor revisions, and I commend the authors on a well-written manuscript.

For section 1.2: “The Caribbean trade of tortoiseshell”, it is my view that the level of detail surrounding the import and export details of tortoiseshell is superfluous. I recommend that this section be revised to be a more concise summary, rather than a detailed explanation, with the caveat that I am not sure how much detail is required by the journal. I will leave this comment to the discretion of the authors and the editors.

Abstract

P1 L26: Specify that tortoiseshell collection is one of the main causes of marine turtle poaching, as it’s currently phrased it makes it sound like this is the only reason for poaching.

P1 L35: Change “protein database” to “protein databases”.

P1 L43: Can remove the de novo sequencing software in the abstract.

Introduction

P2 L8: Change “all around the world” to “globally”.

P2 L42: Add reference for IUCN Red List.

P3 L9: Change “While there is abundance...” to “While there is an abundance...”

P3 L42: Change to “feed the crews onboard vessels...”

P5 L6-7: The last sentence of 1.3 should be moved to the end of the introduction where the author’s outline what was achieved in the study, and set up their hypotheses.

Figure 1. I’m not sure if it’s an issue with the PDF conversion, but parts of Figure 1 are missing (lower left-hand corner). The original uploaded PDF is complete, but I suggest uploading a higher resolution image for final submission.

P8 L12-14: I would hesitate to call flatback turtles “rare.” While they do not occur in the US, or the Caribbean, they are highly abundant in Australia, so I suggest changing this to something along the lines of “geographically localized”.

Methods

P9 L57: Rephrase “delivers” to “possesses” or something similar.

Results

Figure 4. Need more detailed information on what the individual components of the “figure depict”. I’d even argue that this should be included as a table, rather than a figure – as it is effectively a screen grab of a table. For those without a detailed background in proteomics, it would be good to include what the numbers in the table represent (i.e. the length of the residuals), and to explain exactly what peptides A-F actually represent.

P14 L24: Place MAD in parentheses and change “PSMs (peptide-to-spectrum matches)” to “...peptide-to-spectrum matches (PSMs)” for consistency.

Discussion

Generally, well written, with good reasoning and explanation of the findings. I commend the authors for their clear and concise discussion.

Throughout: you use *E. imbricata* and hawksbill turtle interchangeably throughout the discussion. I recommend picking either the common or scientific name and sticking with it throughout to avoid confusion. Similarly, you introduce this species multiple times e.g. P21 L8, only need to use the full scientific name once, with abbreviations to follow.

Decision letter (RSOS-201857.R0)

Dear Dr Solazzo

On behalf of the Editors, we are pleased to inform you that your Manuscript RSOS-201857 "Creation of a peptide database of corneous beta-proteins of marine turtles for the identification of tortoiseshell; Archaeological combs as case study" has been accepted for publication in Royal Society Open Science subject to minor revision in accordance with the referees' reports. Please find the referees' comments along with any feedback from the Editors below my signature.

Please submit your revised manuscript and required files (see below) no later than 7 days from today's (ie 19-Jan-2021) date. Note: the ScholarOne system will 'lock' if submission of the revision is attempted 7 or more days after the deadline. If you do not think you will be able to meet this deadline please contact the editorial office immediately.

on behalf of Professor Matthew Collins (Associate Editor) and Malcolm White (Subject Editor)
openscience@royalsociety.org

Associate Editor Comments to Author (Professor Matthew Collins):

Comments to the Author:

I apologies for the slow response, which was due to waiting for a 3rd reviewer that never provided their review.

In addition to the comments below, I have taken the liberty of suggesting improvements to the text.

Referee #1

This study makes the best use of the available sequence data and generally gives convincing results. However, it is important to emphasize that the identification of species using proteomics is currently limited in power by the lack of the complete genome sequence of *Eretmochelys imbricata* and many other species of turtles. The implications of this problem should be discussed. The authors should also discuss potential improvements that can be expected when complete genome sequences of more turtle species become available, as promised by the Vertebrate Genomes Project.

Figure 6: The legend and the description of this figure in the text are not sufficient to explain the content and to allow interpretation.

Beta-keratins/CBPs are the main proteins of the tortoise shell but there are also other proteins which may be useful for the identification of species. Please discuss this option and state whether or not you have tested or will test this approach.

Minor comments:

Table 3: The tryptic cleavage should have occurred only at the N-terminus of peptide F, not at "N and C". Please check.

Figure 2: Please explain the meaning of letter J in the consensus sequence.

The letter ß does not mean beta, but a special form of "s" in German language. Please change to the real beta symbol β throughout the text.

Page 20 (numbering of the merged pdf file), line 38: Please check if the journal allows highlighting by color in the main text.

Referee #2

"Creation of a peptide database of corneous beta-proteins of marine turtles for the identification of tortoiseshell; Archaeological combs as case study"

General comments

Overall, this is a well written, concise manuscript outlining the proteomic analysis of tortoiseshells, and the application to archaeological identification. I have some general comments that I will briefly touch on here, with more specific comments for each section detailed below. My recommendation is for acceptance with minor revisions, and I commend the authors on a well-written manuscript.

For section 1.2: “The Caribbean trade of tortoiseshell”, it is my view that the level of detail surrounding the import and export details of tortoiseshell is superfluous. I recommend that this section be revised to be a more concise summary, rather than a detailed explanation, with the caveat that I am not sure how much detail is required by the journal. I will leave this comment to the discretion of the authors and the editors.

Abstract

P1 L26: Specify that tortoiseshell collection is one of the main causes of marine turtle poaching, as it’s currently phrased it makes it sound like this is the only reason for poaching.

P1 L35: Change “protein database” to “protein databases”.

P1 L43: Can remove the de novo sequencing software in the abstract.

Introduction

P2 L8: Change “all around the world” to “globally”.

P2 L42: Add reference for IUCN Red List.

P3 L9: Change “While there is abundance...” to “While there is an abundance...”

P3 L42: Change to “feed the crews onboard vessels...”

P5 L6-7: The last sentence of 1.3 should be moved to the end of the introduction where the author’s outline what was achieved in the study, and set up their hypotheses.

Figure 1. I’m not sure if it’s an issue with the PDF conversion, but parts of Figure 1 are missing (lower left-hand corner). The original uploaded PDF is complete, but I suggest uploading a higher resolution image for final submission.

P8 L12-14: I would hesitate to call flatback turtles “rare.” While they do not occur in the US, or the Caribbean, they are highly abundant in Australia, so I suggest changing this to something along the lines of “geographically localized”.

Methods

P9 L57: Rephrase “delivers” to “possesses” or something similar.

Results

Figure 4. Need more detailed information on what the individual components of the “figure depict”. I’d even argue that this should be included as a table, rather than a figure – as it is effectively a screen grab of a table. For those without a detailed background in proteomics, it would be good to include what the numbers in the table represent (i.e. the length of the residuals), and to explain exactly what peptides A-F actually represent.

P14 L24: Place MAD in parentheses and change “PSMs (peptide-to-spectrum matches)” to “...peptide-to-spectrum matches (PSMs)” for consistency.

Discussion

Generally, well written, with good reasoning and explanation of the findings. I commend the authors for their clear and concise discussion.

Throughout: you use *E. imbricata* and hawksbill turtles interchangeably throughout the discussion. I recommend picking either the common or scientific name and sticking with it throughout to avoid confusion. Similarly, you introduce this species multiple times e.g. P21 L8, only need to use the full scientific name once, with abbreviations to follow.

Reviewer comments to Author:

Reviewer: 1

Comments to the Author(s)

This manuscript reports the proteomic analysis of shell samples of turtles and tortoiseshell samples from archeological finds (combs). The authors present peptide sequence data that can be used for the identification of the species from which tortoiseshell was prepared. The data are sound and the manuscript is written well. The results are interesting for specialists in this field. A wider readership will be interested in the general approach of using proteome analysis for research questions outside of classical biology.

Major comments:

This study makes the best use of the available sequence data and generally gives convincing results. However, it is important to emphasize that the identification of species using proteomics is currently limited in power by the lack of the complete genome sequence of *Eretmochelys imbricata* and many other species of turtles. The implications of this problem should be discussed. The authors should also discuss potential improvements that can be expected when complete genome sequences of more turtle species become available, as promised by the Vertebrate Genomes Project.

Figure 6: The legend and the description of this figure in the text are not sufficient to explain the content and to allow interpretation.

Beta-keratins/CBPs are the main proteins of the tortoise shell but there are also other proteins which may be useful for the identification of species. Please discuss this option and state whether or not you have tested or will test this approach.

Minor comments:

Table 3: The tryptic cleavage should have occurred only at the N-terminus of peptide F, not at "N and C". Please check.

Figure 2: Please explain the meaning of letter J in the consensus sequence.

The letter ß does not mean beta, but a special form of "s" in German language. Please change to the real beta symbol β throughout the text.

Page 20 (numbering of the merged pdf file), line 38: Please check if the journal allows highlighting by color in the main text.

Reviewer: 2

Comments to the Author(s)

Review of Solazzo et al. 2020:

"Creation of a peptide database of corneous beta-proteins of marine turtles for the identification of tortoiseshell; Archaeological combs as case study"

General comments

Overall, this is a well written, concise manuscript outlining the proteomic analysis of tortoiseshells, and the application to archaeological identification. I have some general comments that I will briefly touch on here, with more specific comments for each section detailed below. My recommendation is for acceptance with minor revisions, and I commend the authors on a well-written manuscript.

For section 1.2: "The Caribbean trade of tortoiseshell", it is my view that the level of detail surrounding the import and export details of tortoiseshell is superfluous. I recommend that this section be revised to be a more concise summary, rather than a detailed explanation, with the caveat that I am not sure how much detail is required by the journal. I will leave this comment to the discretion of the authors and the editors.

Abstract

P1 L26: Specify that tortoiseshell collection is one of the main causes of marine turtle poaching, as it's currently phrased it makes it sound like this is the only reason for poaching.

P1 L35: Change "protein database" to "protein databases".

P1 L43: Can remove the de novo sequencing software in the abstract.

Introduction

P2 L8: Change "all around the world" to "globally".

P2 L42: Add reference for IUCN Red List.

P3 L9: Change “While there is abundance...” to “While there is an abundance...”

P3 L42: Change to “feed the crews onboard vessels...”

P5 L6-7: The last sentence of 1.3 should be moved to the end of the introduction where the author’s outline what was achieved in the study, and set up their hypotheses.

Figure 1. I’m not sure if it’s an issue with the PDF conversion, but parts of Figure 1 are missing (lower left-hand corner). The original uploaded PDF is complete, but I suggest uploading a higher resolution image for final submission.

P8 L12-14: I would hesitate to call flatback turtles “rare.” While they do not occur in the US, or the Caribbean, they are highly abundant in Australia, so I suggest changing this to something along the lines of “geographically localized”.

Methods

P9 L57: Rephrase “delivers” to “possesses” or something similar.

Results

Figure 4. Need more detailed information on what the individual components of the “figure depict”. I’d even argue that this should be included as a table, rather than a figure – as it is effectively a screen grab of a table. For those without a detailed background in proteomics, it would be good to include what the numbers in the table represent (i.e. the length of the residuals), and to explain exactly what peptides A-F actually represent.

P14 L24: Place MAD in parentheses and change “PSMs (peptide-to-spectrum matches)” to “...peptide-to-spectrum matches (PSMs)” for consistency.

Discussion

Generally, well written, with good reasoning and explanation of the findings. I commend the authors for their clear and concise discussion.

Throughout: you use *E. imbricata* and hawksbill turtle interchangeably throughout the discussion. I recommend picking either the common or scientific name and sticking with it throughout to avoid confusion. Similarly, you introduce this species multiple times e.g. P21 L8, only need to use the full scientific name once, with abbreviations to follow.

===PREPARING YOUR MANUSCRIPT===

If you have been asked to revise the written English in your submission as a condition of publication, you must do so, and you are expected to provide evidence that you have received

language editing support. The journal would prefer that you use a professional language editing service and provide a certificate of editing, but a signed letter from a colleague who is a native speaker of English is acceptable. Note the journal has arranged a number of discounts for authors using professional language editing services (<https://royalsociety.org/journals/authors/benefits/language-editing/>).

===PREPARING YOUR REVISION IN SCHOLARONE===

Author's Response to Decision Letter for (RSOS-201857.R0)

See Appendix A.

Decision letter (RSOS-201857.R1)

Dear Dr Solazzo,

It is a pleasure to accept your manuscript entitled "Creation of a peptide database of corneous beta-proteins of marine turtles for the identification of tortoiseshell; Archaeological combs as case study" in its current form for publication in Royal Society Open Science.

Kind regards,
Royal Society Open Science Editorial Office
Royal Society Open Science

on behalf of Professor Matthew Collins (Associate Editor) and Malcolm White (Subject Editor)
openscience@royalsociety.org

Associate Editor Comments to Author (Professor Matthew Collins):

Associate Editor

Comments to the Author:

Thank you for your rapid response. I think this is a very useful article and congratulate you on your work.

Appendix A

Reviewer comments to Author:

Reviewer: 1

Comments to the Author(s)

This manuscript reports the proteomic analysis of shell samples of turtles and tortoiseshell samples from archeological finds (combs). The authors present peptide sequence data that can be used for the identification of the species from which tortoiseshell was prepared. The data are sound and the manuscript is written well. The results are interesting for specialists in this field. A wider readership will be interested in the general approach of using proteome analysis for research questions outside of classical biology.

Major comments:

This study makes the best use of the available sequence data and generally gives convincing results. However, it is important to emphasize that the identification of species using proteomics is currently limited in power by the lack of the complete genome sequence of *Eretmochelys imbricata* and many other species of turtles. The implications of this problem should be discussed.

This point was made clearer at end of introduction

“Due to the lack of complete genome sequences for marine turtles, proteomics identification is limited to the CBPs sequences of land turtles and a few sequences for the green turtle, making identification of unknown samples to any marine turtle impossible.”

The authors should also discuss potential improvements that can be expected when complete genome sequences of more turtle species become available, as promised by the Vertebrate Genomes Project.

The conclusion was modified to highlight the advantages of using de novo sequencing in the absence of genomes as these might still take a few years to come, but also the increase in confidence expected from having genome sequences.

“As species get sequenced in coming years, it will add confidence in the identification of the markers established from de novo sequencing and resolve issues of sequence homology. In the meantime, this study has showed that automated and targeted de novo sequencing could be a powerful tool to characterize unknown proteins in species without a sequenced genome, and a faster approach than DNA sequencing.”

Figure 6: The legend and the description of this figure in the text are not sufficient to explain the content and to allow interpretation.

We added the following to clarify: “The k-means SOM clusters were found to represent the following species: 1) *C. caretta*, 2) *C. mydas/L. kempji*, 3) *L. olivacea*, and 4) *E. imbricata*. Based on these clusters, the combs were identified as deriving from *E. imbricata* (SP1074, SP6554, SP88, SP97), *L. olivacea* (Inv16914), or *C. mydas/L. kempji* (SP6536, TAB002) (Figure 6a).”

and to the figure caption: “Figure 6: PCA analysis of each sample with a) SOM clustering or b) species/archaeological type. The k-means SOM clusters represent the following species: 1) *C. caretta*, 2) *C. mydas/L. kempji*, 3) *L. olivacea*, and 4) *E. imbricata*”

Beta-keratins/CBPs are the main proteins of the tortoise shell but there are also other proteins which may be useful for the identification of species. Please discuss this option and state whether or not you have tested or will test this approach.

The data on alpha-keratins were not included in this paper for several reasons: they were minor proteins, they didn't seem to have as much variability in sequences as the CBPs (few de novo peptides were found), and they are mostly absent from the archaeological samples. Although they are interesting in the context of tortoiseshell degradation, they did not add much in terms of species identification which was the main focus of this study. Adding these data will probably overload the paper.

These points are now briefly explained across the paper.

Minor comments:

Table 3: The tryptic cleavage should have occurred only at the N-terminus of peptide F, not at “N and C”. Please check.

Corrected

Figure 2: Please explain the meaning of letter J in the consensus sequence.

J represents both leucine and isoleucine. As isomers they have the same mass and can't be differentiated. Sentence added in figure caption: “X indicates a residue where there is no consensus and J stands for the isomers leucine L and isoleucine I.”

The letter β does not mean beta, but a special form of "s" in German language. Please change to the real beta symbol β throughout the text.

Thanks for noticing. Corrected.

Page 20 (numbering of the merged pdf file), line 38: Please check if the journal allows highlighting by color in the main text.

I could not find that information

Reviewer: 2

Comments to the Author(s)

Review of Solazzo et al. 2020:

"Creation of a peptide database of corneous beta-proteins of marine turtles for the identification of tortoiseshell; Archaeological combs as case study"

General comments

Overall, this is a well written, concise manuscript outlining the proteomic analysis of tortoiseshells, and the application to archaeological identification. I have some general comments that I will briefly touch on here, with more specific comments for each section detailed below. My recommendation is for acceptance with minor revisions, and I commend the authors on a well-written manuscript.

For section 1.2: "The Caribbean trade of tortoiseshell", it is my view that the level of detail surrounding the import and export details of tortoiseshell is superfluous. I recommend that this section be revised to be a more concise summary, rather than a detailed explanation, with the caveat that I am not sure how much detail is required by the journal. I will leave this comment to the discretion of the authors and the editors.

See proposed cuts in 1.2

Abstract

P1 L26: Specify that tortoiseshell collection is one of the main causes of marine turtle poaching, as it's currently phrased it makes it sound like this is the only reason for poaching.

Corrected to "remains one of the main causes"

P1 L35: Change "protein database" to "protein databases".

Corrected

P1 L43: Can remove the de novo sequencing software in the abstract.

Corrected

Introduction

P2 L8: Change "all around the world" to "globally".

Corrected

P2 L42: Add reference for IUCN Red List.

Added

P3 L9: Change "While there is abundance..." to "While there is an abundance..."

Corrected

P3 L42: Change to "feed the crews onboard vessels..."

Corrected

P5 L6-7: The last sentence of 1.3 should be moved to the end of the introduction where the author's outline what was achieved in the study, and set up their hypotheses.

Moved

Figure 1. I'm not sure if it's an issue with the PDF conversion, but parts of Figure 1 are missing (lower left-hand corner). The original uploaded PDF is complete, but I suggest uploading a higher resolution image for final submission.

Happens when converting into pdf.

P8 L12-14: I would hesitate to call flatback turtles "rare." While they do not occur in the US, or the

Caribbean, they are highly abundant in Australia, so I suggest changing this to something along the lines of "geographically localized".

Sentence changed to "The flatback turtle, geographically confined to Australia, was not included in this study as no scute samples were available at NMNH"

Methods

P9 L57: Rephrase "delivers" to "possesses" or something similar.

The sentences were merged into "Found under the skull of the deceased female (18 to 24 year old), this curved, 11.7 cm long comb has a single row of 14 teeth widely spaced, the majority of which are preserved."

Results

Figure 4. Need more detailed information on what the individual components of the "figure depict". I'd even argue that this should be included as a table, rather than a figure – as it is effectively a screen grab of a table. For those without a detailed background in proteomics, it would be good to include what the numbers in the table represent (i.e. the length of the residuals), and to explain exactly what peptides A-F actually represent.

The figure was modified to more closely resemble an actual figure, and the caption was modified

P14 L24: Place MAD in parentheses and change "PSMs (peptide-to-spectrum matches)" to "...peptide-to-spectrum matches (PSMs)" for consistency.

Corrected

Discussion

Generally, well written, with good reasoning and explanation of the findings. I commend the authors for their clear and concise discussion.

Throughout: you use *E. imbricata* and hawksbill turtle interchangeably throughout the discussion. I recommend picking either the common or scientific name and sticking with it throughout to avoid confusion. Similarly, you introduce this species multiple times e.g. P21 L8, only need to use the full scientific name once, with abbreviations to follow.

This was changed to introduce the names in paragraph 1.1, and the scientific names are now used in abbreviated form throughout the results and discussion